

Simultaneous Measurement of Urban and Rural Particles in Beijing, Part II: Case Studies
of Haze Events and Regional Transport
Yang Chen,[1] Guangming Shi,[1,3] Jing Cai,[2] Zongbo Shi,[4,5] Zhichao Wang,[1] Xiaojiang Yao,[1]
Mi Tian,[1] Chao Peng,[1] Yiqun Han,[2] Tong Zhu,[2] Yue Liu,[2] Xi Yang,[2] Mei Zheng,[2*] Fumo
Yang,[1,3*] and Kebin He[6]
[1] Chongqing Institute of Green and Intelligent Technology, Chinese Academy of Sciences,
Chongqing 400714, China
[2] SKL-ESPC and BIC-ESAT, College of Environmental Sciences and Engineering, Peking
University, Beijing 100871, China
[3] Department of Environmental Science and Engineering, College of Architecture and
Environment, Sichuan University, Chengdu 610065, China
[4] School of Geography, Earth and Environmental Sciences, the University of Birmingham,
Birmingham B15 2TT, UK
[5] Institute of Surface-Earth System Science, Tianjin University, Tianjin 300072, China
[6] School of Environment, Tsinghua University, Beijing 100084, China
Corresponding    to    Fumo    Yang    (fmyang@scu.edu.cn)    and    Mei    Zheng
(mzheng@pku.edu.cn)



Abstract
Two parallel field studies were conducted simultaneously at both urban and rural sites in
Beijing from 11/01/2016 to 11/29/2016. Online single-particle chemical composition
analysis was used as a tracer system to investigate the impact of heating activities and
formation of haze events. Central heating elevated EC-Nit, EC-Nit-Sul, and ECOC-Nit
levels by 1.5–2.0 times due to the increased use of coal in the urban areas. However, in the
rural areas, residential heating which mainly consumes low-quality coal and biomass
burning elevated ECOC-Nit-Sul, Nak-Nit, and OC-Sul levels by 1.2–1.5 times. Four severe
haze events (hourly $PM_{2.5} > 200$ µg m$^{-3}$) occurred at both sites during the studies. In each
event, a pattern of "transport and accumulation" was found. In the first stage, particles were
regionally transported from the south or southwest and accumulated under air stagnations,
creating significant secondary formation. Consequently, the boosting of $PM_{2.5}$ led to severe
haze. At both sites, the severe haze occurred due to different patterns of local emission,
transport, and secondary processes. At PG, the sulfate-rich residential coal burning
particles were dominant. The regional transport between PG and PKU was simulated using
the WRF-HYSPLIT model, confirming that the transport from PG to PKU was significant,
but PKU to PG occurred occasionally. These cases can explain the serious air pollution in
the urban areas of Beijing and the interaction between urban and rural areas. This study
can provide references for enhancing our understanding of haze formation in Beijing.
Keywords: urban; regional; single particle; transport; pollution event





## 1. Introduction

The Beijing-Tianjin-Hebei (BTH) area in China has been suffering from extreme haze events caused by high concentrations of $PM_{2.5}$ ($> 200\,\mu g\,m^{-3}$) since 2013 (Guo et al., 2014). Studies have been performed to understand the formation of such massive haze events in Beijing (Tian et al., 2014; Quan et al., 2013; Che et al., 2014; He et al., 2015). Traffic, cooking, and coal combustion emissions accounted for 41–59% of the total submicron organic aerosols, and the remainder were secondary organic aerosols (Sun et al., 2014). Model studies suggest that temperature inversion, low boundary layer, and transported pollutants cause the local accumulation of $PM_{2.5}$ in urban areas (Zhang et al., 2015). In short, significant local emissions, unfavorable meteorological conditions, and regional transport play essential roles in accumulating $PM_{2.5}$.

There are unresolved issues surrounding whether the rapid boosting of PM in Beijing is due to local secondary aerosol formation or transport. Wang et al. (2016) have proposed that the accumulation of nitrates is dominant at the beginning of haze events, and then sulfate increases because $SO_2$ is oxidized into sulfate in ammonium-rich conditions. Moreover, Cheng et al. (2016) have suggested that $NO_2$ could oxidize $SO_2$ to sulfate on the surface of alkali aerosols. However, Li et al. (2015) have argued that regionally transported $PM_{2.5}$ is a significant cause of severe haze. Last but not least, Sun et al. (2014); Sun et al. (2013a) have proposed that both local formation and regional transport are factors. Except for model studies, most field studies have focused on urban areas in Beijing, with limited attention to rural areas. The characterization of rural PM is also essential to understanding the evolution of particulate haze events.





The cold winter results in the necessity of heating, consequently impacting the air quality
in BTH (Sun et al., 2014). In urban areas, central heating systems use coal or natural gas,
while rural households use coal or biofuel for heating and cooking. Residential emissions
in Beijing reach about 4 million tons, mainly caused by low-efficiency coal combustion
(Li et al., 2015). Coal combustion organic aerosols (CCOA) account for 20–32% of total
submicron OA in Beijing (Sun et al., 2014; Sun et al., 2013a). However, whether CCOA
is contributed by central or household heating remains unclear. Notably, central and
household heating release distinct particles due to different burning conditions (Lee et al.,
2005; Chagger et al., 1999). Therefore, analyzing household heating and cooking emissions
in rural areas is also beneficial for understanding the source of urban $PM_{2.5}$ in Beijing.
As mentioned in Part I (Chen et al., 2020), two SPAMSs were deployed simultaneously in
Peking University (PKU) and Pinggu (PG) in order to monitor urban and rural particles in
the Beijing region. In Part II, the detailed analysis of haze events, effects of heating
activities, and evidence of regional transport between urban and rural areas are addressed.
**2. Methodology**
**2.1 Sampling sites, instrumentation, and data analysis**
Please refer to Part I and Support Information for the detail (Chen et al., 2020). Briefly, the
field studies were performed simultaneously at PKU (116.32ºE, 39.99ºN) and PG (117.05
ºE, 40.17ºN) from 11/01/2016 to 11/29/2016. The two sites represent both typical urban
and rural areas, respectively.  The local meteorological data is retrieved from the local
meteorological offices. Two SPAMSs (0515, Hexin Inc., Guangzhou, China) were



deployed at both sites for parallel measurements. SPAMS generates single particle mass
spectra from the captured individual particles. The technical description of SPAMS is
available in the literature (Li et al., 2011). A neural network algorithm based on adaptive
resonance theory (ART-2a) was applied for clustering particle types in the datasets (Song
et al., 1999). During the clustering procedure, the relative peak areas (RPA) of sulfate and
nitrate are considered. A criterion of RPA >0.1 is used to identify the nitrate-rich (-Nit),
sulfate-rich (-Sul), or both. Based on the strategy, 20 and 19 particle types were identified
at PKU and PG respectively.
**2.2 Dispersion model**
A WRF-HYSPLIT (Weather Research and Forecasting - Hybrid Single Particle
Lagrangian Integrated Trajectory) coupling model was used to describe the air parcel
movement between PKU and PG. The description of the model is available at
https://www.arl.noaa.gov/hysplit/inline-wrf-hysplit-coupling/. The HYSPLIT dispersion
simulations were driven by the meteorological data fields from the WRF model version
3.8. The WRF domains are shown in Figure 2. The innermost domain was configured to
cover northern China with a horizontal resolution of 3 km and 35 vertical layers. The
longwave and shortwave radiation schemes were set as the RRTMG and Dudhia scheme
respectively. The Yonsei University (YSU) scheme was used for the PBL parameterization.
For the microphysics, the Morrison 2-moment scheme was adopted. NCEP FNL (National
Centers for Environmental Prediction, final) data with a resolution of 1°×1° was employed
as initial and boundary conditions. The WRF simulation was initialized as a "cold start" at
0000 UTC each day and ran for 36 hours. The first 12 hours were discarded as model spin-
up time, and the output for the following 24 hours was retained. This process was repeated
to produce continuous meteorological data fields for the whole experimental period. The
HYSPLIT was set to release 10,000 Lagrangian particles within one hour at PKU and PG,
10 m above ground level. The concentration of released particles was simulated with one
vertical layer extending from 0 to 1,000 m above ground level.

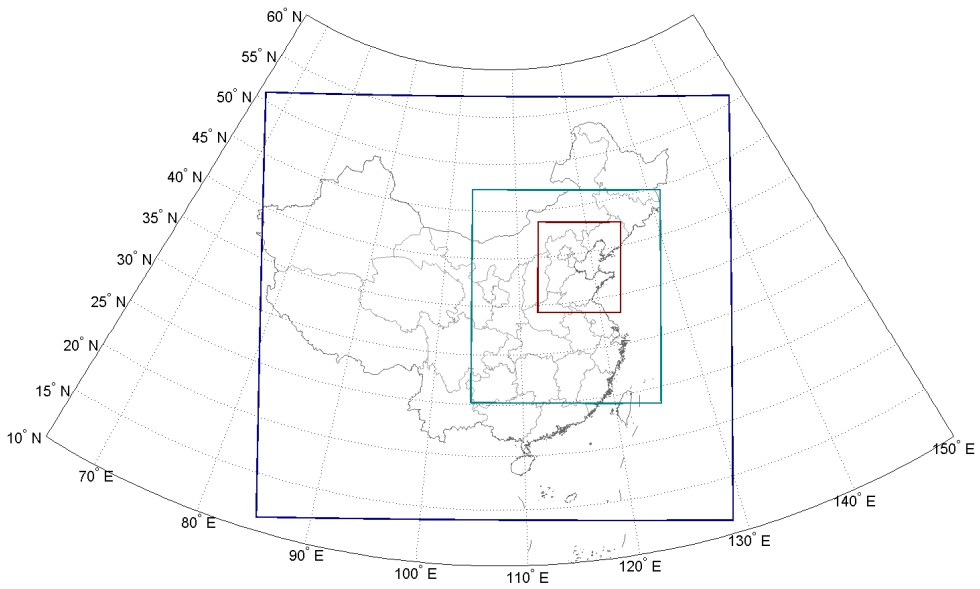


Figure 1. Spatial configuration of domains used for WRF simulation.
**3. Results and discussion**
**3.1 Particle type description**
Particle types, their ratios at both sites, and major chemical composition are shown in Table
1. The typical single-particle mass spectra of each particle type are available in Supportive
Information and (Chen et al., 2020).





Table 1. Particle types and their relative ratios and chemical composition

| | Both | PKU | PG | Chemical Composition* |
|---|---|---|---|---|
| EC | EC-Nit | 7.0 | 2.0 | $C_n^+$, $C_n^-$, $HSO_4^-$, $NO_2^-$, |
| | EC-Nit-Sul | 10.5 | 3.5 | $NO_3^-$ |
| | EC-Sul | 0.7 | 0.1 | |
| ECOC | ECOC-Nit-Sul | 12.0 | 18.6 | $C_n^+$, $C_n^-$, $C_xH_y^+$, $C_xH_yO_z^+$ |
| | ECOC-Sul | 12.7 | 9.8 | $HSO_4^-$, $NO_3^-$ |
| K-rich | K-rich | 7.2 | 6.4 | $K^+$, $NH_4^+$, $HSO_4^-$, $NO_3^-$ |
| | K-Nit | 8.0 | 8.2 | $NO_2^-$ |
| | K-Nit-Sul | 16.0 | 1.9 | |
| | K-Sul | 0.6 | 4.5 | |
| NaK | NaK | 0.4 | 1.8 | $Na^+$, $K^+$, $NH_4^+$, $HSO_4^-$, |
| | NaK-Nit | 6.4 | 1.7 | $NO_3^-$ |
| | NaK-Nit-Sul | 2.5 | 1.9 | |
| | NaK-Sul | 0.2 | 0.4 | |
| OC | OC-Nit-Sul | 7.4 | 21.3 | $C_xH_y^+$, $C_xH_yO_z^+$, $NH_4^+$ |
| | OC-Sul | 0.9 | 6.9 | $HSO_4^-$, $NO_3^-$ |
| | Ca-dust | 0.4 | 0.1 | $Cl^-$ |
| Fe | Fe-rich | 3.1 | 1.8 | $Fe^+$, Org, $HSO_4^-$, $NO_3^-$ |
| | ECOC-Nit | 3.1% | | |
| | OC-Nit | 0.9% | | |
| | K-Amine-Nit-Sul | 0.1% | | TMA, $NH_4^+$, $HSO_4^-$, $NO_3^-$ |
| | ECOC | | 5.9% | $C_n^+$, $C_n^-$, $C_xH_y^+$, $C_xH_yO_z$ |
| | OC | | 3.3% | $C_xH_y^+$, $C_xH_yO_z$ |

* chemical species with relative peak area >0.1

**3.2 Overview of haze events**

Figures 2 and 3 show the overview of PM$_{2.5}$, meteorology parameters, and time trends of
particles at PKU and PG respectively. There were four parallel haze events during the
observation period: 11/01/2016–11/07/2016 (E1), 11/09/2016–11/15/2016 (E2),
11/15/2016–11/22/2016 (E3), and 11/25/2016–11/28/2016 (E4).

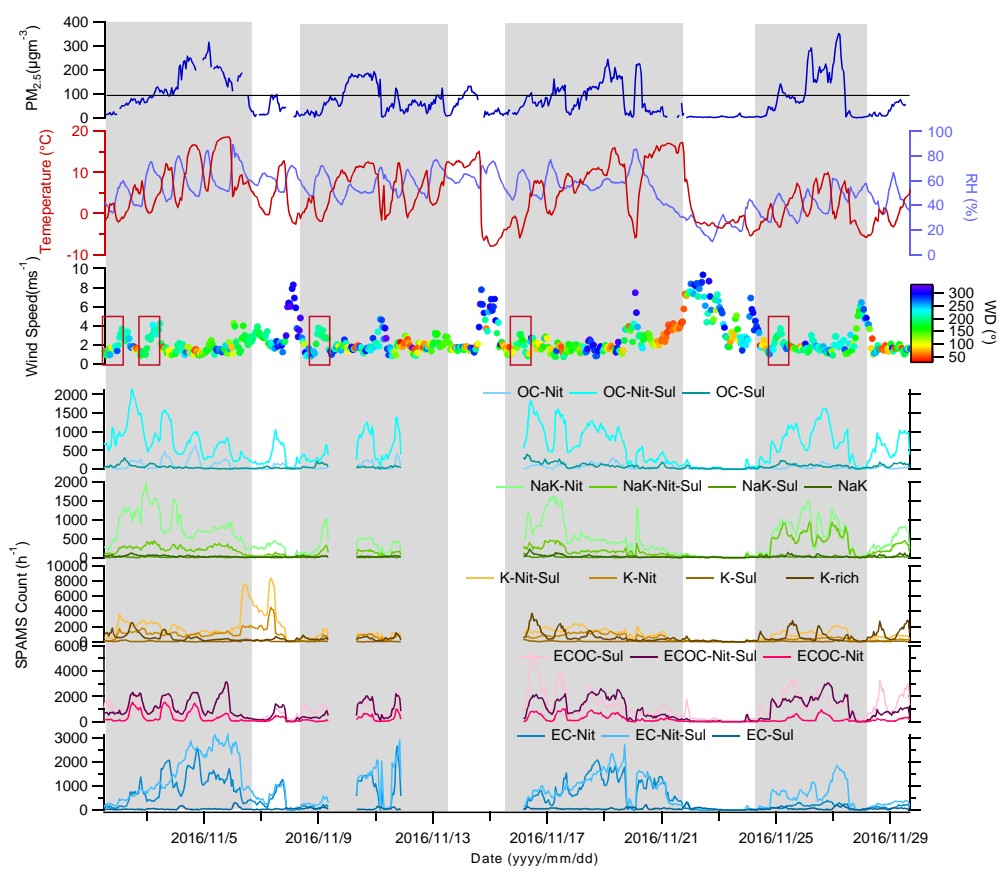


Figure. 2. Time trends of PM$_{2.5}$, temperature, relative humidity, wind direction, wind speed,
and single particle types at PKU. The rectangles indicate the transport of regional particles.

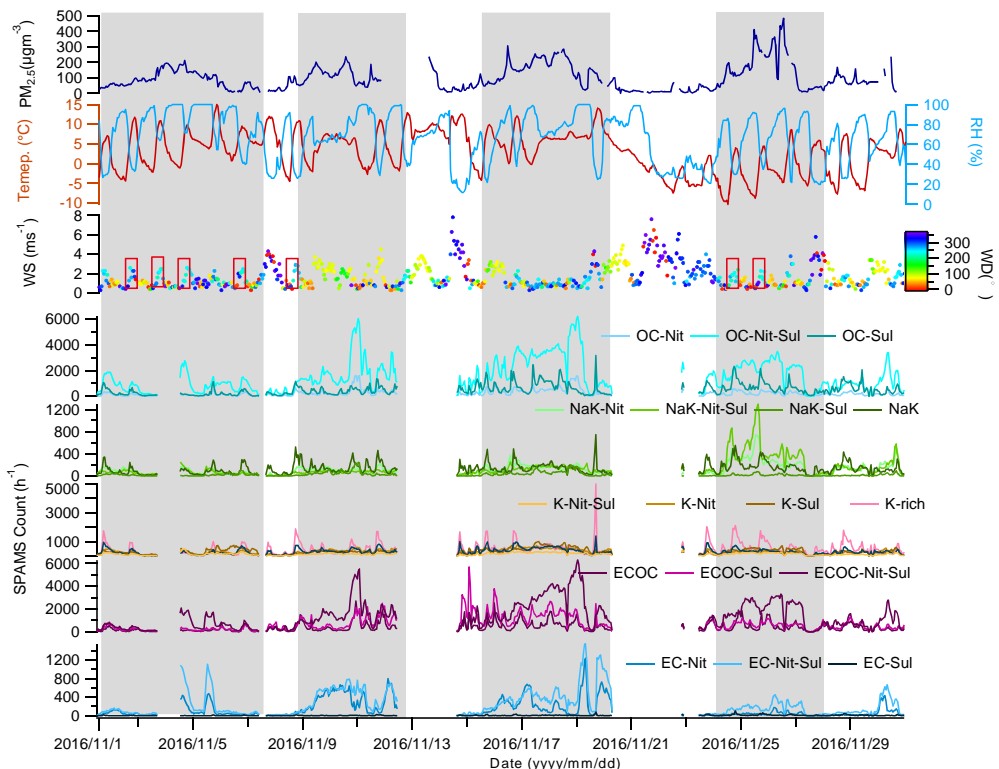


Figure 3. Time trends of PM2.5, temperature, relative humidity, wind direction, wind speed, and single particle types at PG. The rectangles indicate the transport of regional particles.

The pattern of single-particle chemical composition, represented by normalized number
fractions of particle types in different periods, is used to describe PM characteristics. The
correlations of normalized number fractions during events at PKU and PG are shown in
Tables 1 and S3. E1_PKU was well correlated with Clear1 ($R$ = 0.90) and E2_PKU ($R$ =
0.86), but poorly correlated with Clear2 ($R$ = 0.38) and E4 ($R$ = 0.64). This is because
E1_PKU and E2_PKU occurred before the heating period, but E4_PKU occurred after
(11/15/2016). The chemical compositions of the four events at PG are highly correlated





with each other (all $R$s > 0.90, Table S3). These results indicate that the chemical
composition patterns changed significantly at PKU, but insignificantly at PG.
Table 1. Correlations of number fractions of particle types in different events at PKU.

|  | E1 | Clear1 | E2 | Clear2* | E4 |
|---|---|---|---|---|---|
| E1 | 1 |  |  |  |  |
| Clear1 | 0.90 | 1 |  |  |  |
| E2 | 0.86 | 0.91 | 1 |  |  |
| Clear2 | 0.38 | 0.70 | 0.58 | 1 |  |
| E4 | 0.64 | 0.81 | 0.83 | 0.76 | 1 |

Note: The chemical composition of E3 is unavailable.

**3.3 Influence of heating activities**

Central heating began on 11/15/2016 in the urban area, while residential heating in the rural
area had no distinct starting day. As such, the shift in emissions due to the increased use of
solid fuel directly affected the particulate chemical composition. As shown in Figure 4, the
normalized ratios of EC-Nit_PKU, EC-Nit-Sul_PKU, and OC-Nit_PKU increased by
about 1.5 times. EC-Nit_PKU and EC-Nit-Sul_PKU came from multiple local sources, one
of which was coal burning in boilers (Xu et al., 2018b). In addition, high EC concentrations
have been observed during the heating period each year for decades (Chen et al., 2016b).
The mass spectra of OC-Nit particles were composed of a series of ion fragments of
polycyclic aromatic hydrocarbons (PAHs). The results are consistent with organic aerosols
from coal burning in AMS-related studies (Wang et al., 2019; Sun et al., 2013b).
Additionally, $PM_{2.5}$-bound PAHs increased by three times when the heating period began
in Beijing (Zhang et al., 2017). The results also suggest the potential health risks of coal
burning in wintertime in Beijing (Linak et al., 2007; Chen et al., 2013).



Biomass burning (BB) has been proven as a significant source of $PM_{2.5}$ in Beijing (Sun et
al., 2013b; Sun et al., 2014), accounting for 9–12% (Liu et al., 2019). Anthropogenic BB,
e.g. burning household biofuel, is prohibited in urban areas, but common in the areas
surrounding Beijing. Most BB-related particles such as K-rich, K-Nit, and K-Nit-Sul at
PKU were regional (Part I)(Chen et al., 2020). Not surprisingly, K-Nit_PKU and K-Nit-
Sul_PKU both increased to 1.7 times after 11/15/2016. Interestingly, K-Amine-Nit_PKU
increased by 2.3 times after the heating period began, suggesting that BB is also a source
of particulate amines in Beijing (Chen, 2019).
After 11/15/2016, NaK-Nit-Sul_PG, Ca-rich_PG, and OC-Sul_PG increased by 1.96, 1.30,
and 1.47 times respectively. As described above, in rural areas, low-quality coal is
commonly used for heating and cooking, resulting in abundant EC-Sul, OC-Sul, and NaK-
Nit-Sul (Xu et al., 2018a; Chen et al., 2016a). Interestingly, Ca-rich particles that were well
correlated with OC-Sul ($R = 0.79$) also increased, possibly due to flying ash from coal
stoves.
A number of studies have reported contributions of coal burning to the submicron PM in
urban areas of Beijing. According to these mass-based studies, PM-bound PAHs, chloride,
sulfate, nitrate, and lead were markers from emissions of coal burning (Xu et al., 2018a;
Sun et al., 2014; Ma et al., 2016; Zhang et al., 2019). Our result shows that these species
were internally mixed as the ECOC particles. In particular, the household heating in PG
released significant fractions of ECOC particles that arrived in the urban areas of Beijing.
Likewise,   K-rich particles from BB also transport to the urban areas of Beijing.





Conclusively, control of emissions from household emissions is also a key to improve the
air quality in Beijing




Figure 4. Variation of particle number ratio at PKU and PG before and after the heating
period 2017.





### 3.4 Case studies: Haze events at PKU

As shown in Figure 2, before $PM_{2.5}$ increased to 100 µg m$^{-3}$ during E1_PKU, two processes

of $PM_{2.5}$ transport were observed. The first process was from 12:00 on 11/01/2016 to 2:00

on 11/02/2016, in which OC-Nit-Sul, K-Nit-Sul, K-Nit, NaK-Nit, K-Nit-Sul increased

dramatically as the southern wind speed increased from 1.3 m s$^{-1}$ to 3.7 m s$^{-1}$. The wind

speed then decreased to 1.2 m s$^{-1}$ until 16:00 on 11/02/2016, and the accumulation of $PM_{2.5}$

resulted in a concentration of 67 µg m$^{-3}$. The second process occurred from 17:00 on

11/02/2016 to 16:00 on 11/03/2016. Severe accumulation then started at 1:00 on

11/04/2016, with an elevating trend of RH, reaching the highest $PM_{2.5}$ level of 314 µg m$^{-3}$

at 03:00 on 11/05/2016. After that, the wind dispersed the $PM_{2.5}$ to 11 µg m$^{-3}$ at 17:00 on

11/06/2016. In short, regional particles were transported from the south or southwest, then

the accumulation of $PM_{2.5}$ began. The accumulation of pollutants was accompanied by

secondary aerosol formation, causing severe haze events.

During the events at PKU (Figure 2), particles transported from the south and southwest

were observed and labeled with red rectangles. During E4_PKU, the $PM_{2.5}$ concentration

increased from 6 µg m$^{-3}$ to 122 µg m$^{-3}$ between 15:00 on 11/24/2016 and 3:00 on

11/25/2016 due to the southern wind, which brought abundant NaK-Nit, NaK-Nit-Sul,

ECOC-Nit-Sul, and EC-Nit-Sul. Notably, regional particles were dramatically different

from those of E1_PKU due to the heating period. Then, under stagnant air conditions, the

accumulation began at 22:00 on 11/25/2016 and lasted until 03:00 on 11/26/2016, with

$PM_{2.5}$ levels reaching 281 µg m$^{-3}$. At this stage, such local particles as OC-Nit-Sul, ECOC-

Nit-Sul, and ECOC-Nit also showed accumulation and local emissions, while both the K-

rich and NaK families showed a pattern of transport and accumulation (Figures 5 and 6).





As shown in Figure 5, which gives an integrated view of related particle types in urban
Beijing, three types of particle evolution are distinguished during E1. First, EC particles,
including EC-Nit, EC-Nit-Sul, and EC-Sul, show trends of accumulation, but with clear
patterns of emissions, suggesting a pattern of emission and accumulation. Second, for
regional particles such as the K-rich and NaK families, the processes of transport and
accumulation were identified, with significant accumulation but unclear diurnal patterns.
Third, the OC and ECOC families illustrated clear diurnal patterns of local emission and
evolution. Notably, during the development of E1, the ratio of aged ECOC-Nit-Sul
increased from 20% to 83%, suggesting that significant secondary processing occurred.
Due to the nature of SPAMS, the quantitative measurement of secondary formation is
unavailable. Fortunately, as an integrated and extensive project, APHH-Beijing also
included the online monitoring of the chemical composition of $PM_{2.5}$. For example, during
the transport stage of E4_PKU, $PM_{2.5}$ was composed of 60% organic matter (OM) and 40%
total nitrate, sulfate, and ammonium. During the accumulation stage, sulfate, nitrate, and
ammonium levels were boosted up to 123 µg m$^{-3}$ (63%) together (Liu et al., 2019). Wang
et al. (2019) also reported that, during the accumulation stage of E4_PKU, the elevation of
secondary OOA1 and OOA2 was significant.



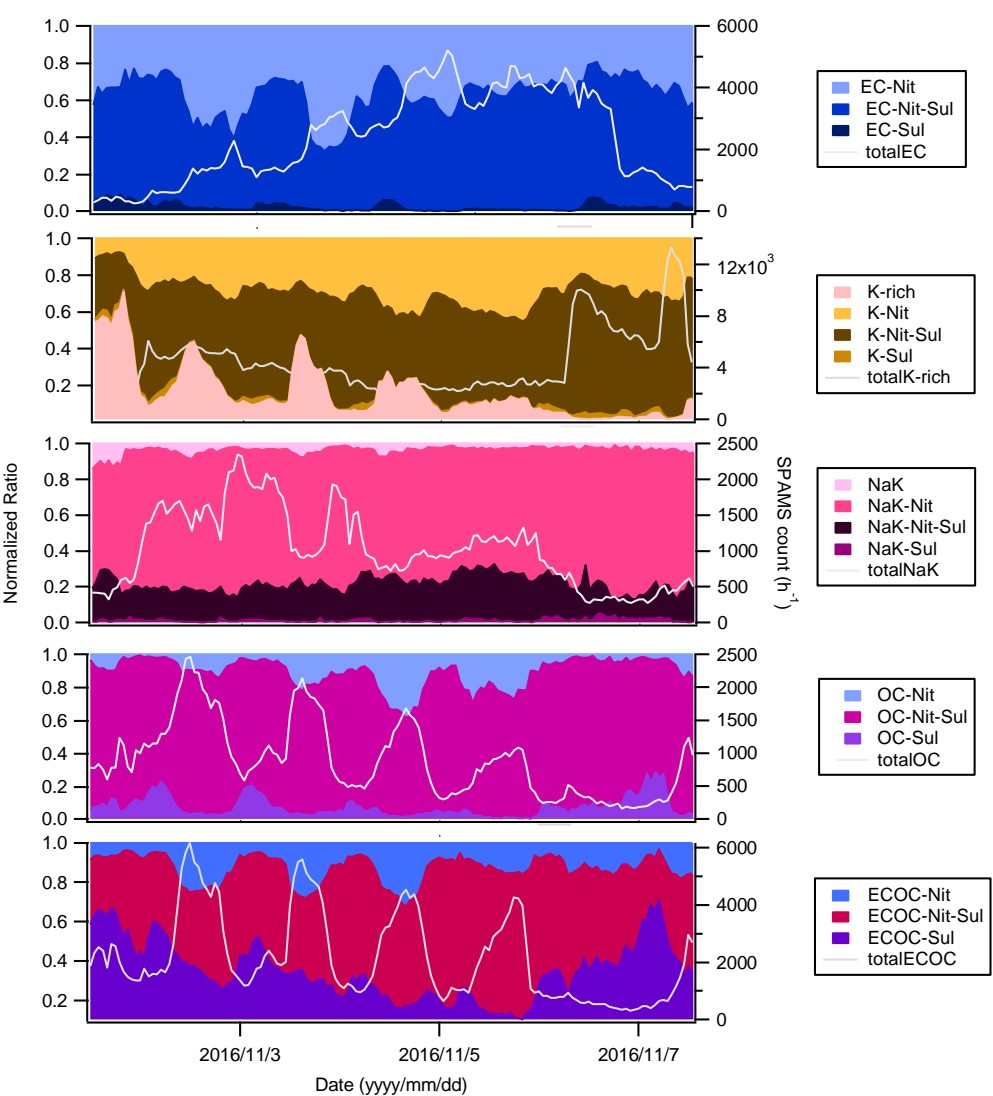

Figure 5. Time trends of number ratios of particle types (left) and hourly counts of particle families (EC, BB, NaK, OC, and ECOC, right) during Pollution Event 1 (E1 11/01–11/08) at PKU.



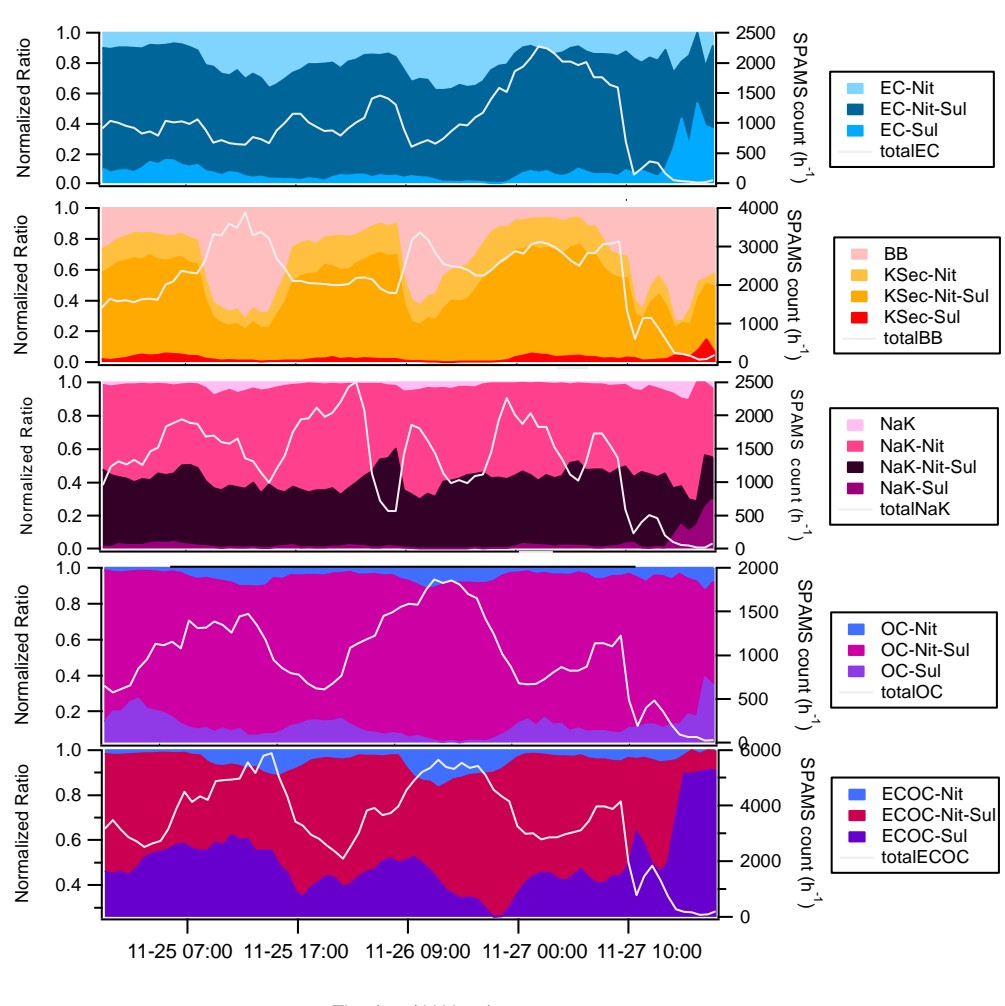


Time(mm/dd hh:00)

Figure 6. Time trends of number ratios of particle types (left) and hourly counts of particle
families (EC, BB, NaK, OC, and ECOC, right) during Pollution Event 4 (E4) at PKU.





### 3.5 Case studies: Haze events at PG


A pollution event occurred at PG (E1_PG) from 11/01 to 11/08. During this period, a
similar pattern of transport and accumulation was also observed. At the beginning of each
pollution event, there was also a transport process of particles from the southwest (Figure
3); when the wind speed reached $< 2$ m s$^{-1}$, accumulations began, and the haze dispersed
with the elevating wind speed. The development of haze events was similar, and Figure 3
lists all the favorable wind directions for transport with red rectangles. As shown in Figure
8, EC-Nit and EC-Nit-Sul showed unclear diurnal patterns, indicating that both particle
types were transported regionally. K-rich, NaK, OC, and ECOC had clear diurnal heating
and cooking patterns, suggesting that local sources were dominant. Such aged particle
types as OC-Nit-Sul and ECOC-Nit-Sul increased due to local aging processes during
E1_PG. Therefore, E1_PG was mainly driven by the input of particles, local emissions,
and accumulation. Moreover, the relative abundance of ECOC-Nit-Sul increased twofold
from 2:00 on 11/03/2016 to 12:00 on 11/03/2016, suggesting the contribution of secondary
formation (Figure 8).
When E4_PG occurred, transport from the southwest was identified along with the
transport of EC-Sul and EC-Nit-Sul, resulting in a PM$_{2.5}$ concentration of 176 µg m$^{-3}$ at
10:00 on 11/24/2016. The average wind speed was 1.5 ms$^{-1}$ at the time, representing a
typical stagnant-air condition. All particle families showed accumulation trends after that
(Figure 3). The sharp decrease of all particle families was due to the high western wind
speed ($> 4$ ms$^{-1}$) at 12:00 on 11/26/2016. During particulate accumulation at PG, such local
particle types as ECOC, OC, and NaK still had diurnal patterns, but the aged "-Nit-Sul"




particles types were predominant (> 50% in all particle families). Thus, the local
accumulation of pollutants was the major driver of E4_PG (Figure 8).

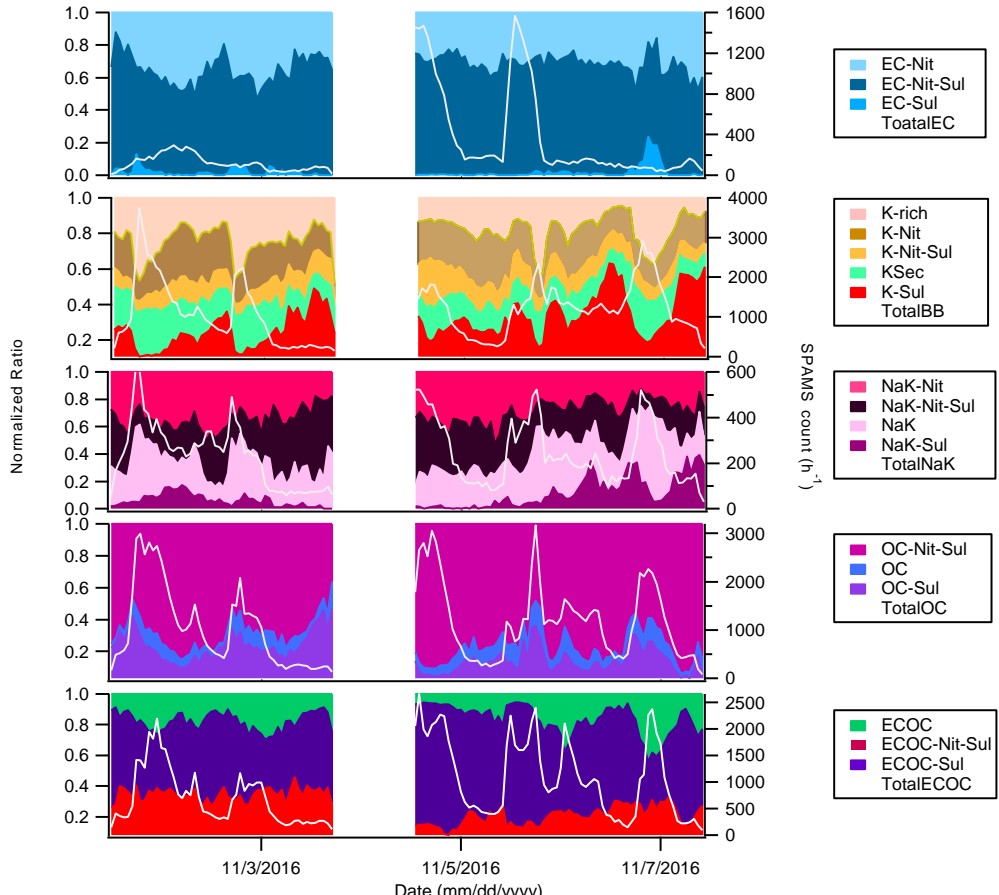


Figure 7. Time trends of number ratios of particle types (left) and hourly counts of particle

families (EC, BB, NaK, OC, and ECOC, right) during Pollution Event 1 (E1 11/01–11/08)

at PG.

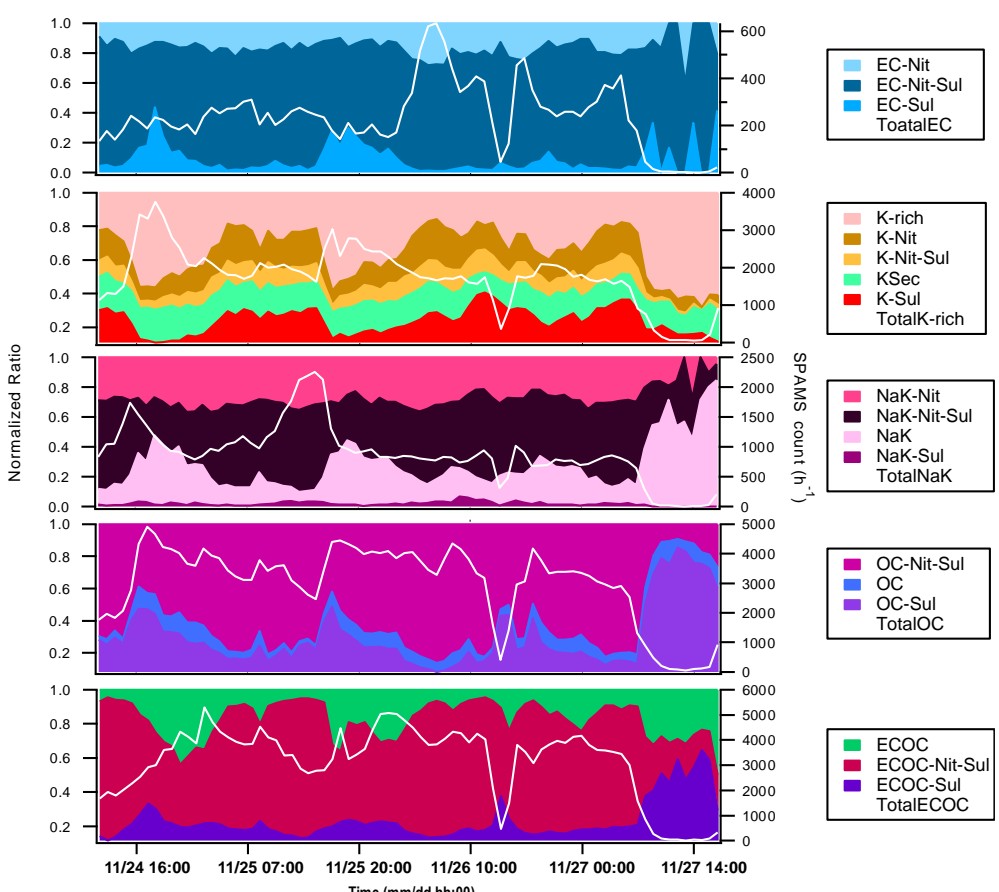


Figure 8. Time trends of number ratios of particle types (left) and hourly counts of particle

families (EC, BB, NaK, OC, and ECOC, right) during Pollution Event 4 (E4) at PG.

**3.6 Interaction of PM between PKU and PG**

Since PKU and PG share 17 common particle types, possible transport between the two

sites was validated using the HYSPLIT model. All cases of transport are available in

Supplementary information (Figures S11 and S12). Figures 9 and 10 only illustrate the

examples of transport during each pollution event. The PKU site is located on the edge of

plumes originating from PG during E1, which implies that the particulate transport was





partially from PG (Figure 9). Moreover, the PKU site lies in the high concentration zone
of plumes PG from during E3 and E4. Therefore, E3_PKU and E4_PKU were confidently
considered input haze events. In contrast, the relatively slighter transport of air mass from
PKU to PG was observed during these events. As shown in Figure 10, the air mass passing
through the PKU site mainly influenced the areas in the south and east. Consequently, the
PG site was seldom in the high concentration zone of plumes originating from PKU.
Figures 9 and 10 suggest that pollutants were transported significantly from PG to PKU
during stagnant air conditions when dense haze occurred. These results are consistent with
the analysis of particle categories. In an urban area such as PKU, the local EC particles
were associated with the ECOC and OC families causing severe pollution in the urban area.
On the other hand, in the rural area, the aged particles were dominant under stagnant air
conditions and transported to PKU, leading to extreme urban particulate pollution. Besides,
our results are consistent with other studies in the APHH-Beijing Project. For example, Du
et al. (2019) have confirmed that regional transport plays a non-negligible role in haze
episodes with contributions of 14–31% to the surface $PM_{2.5}$ mass concentration.



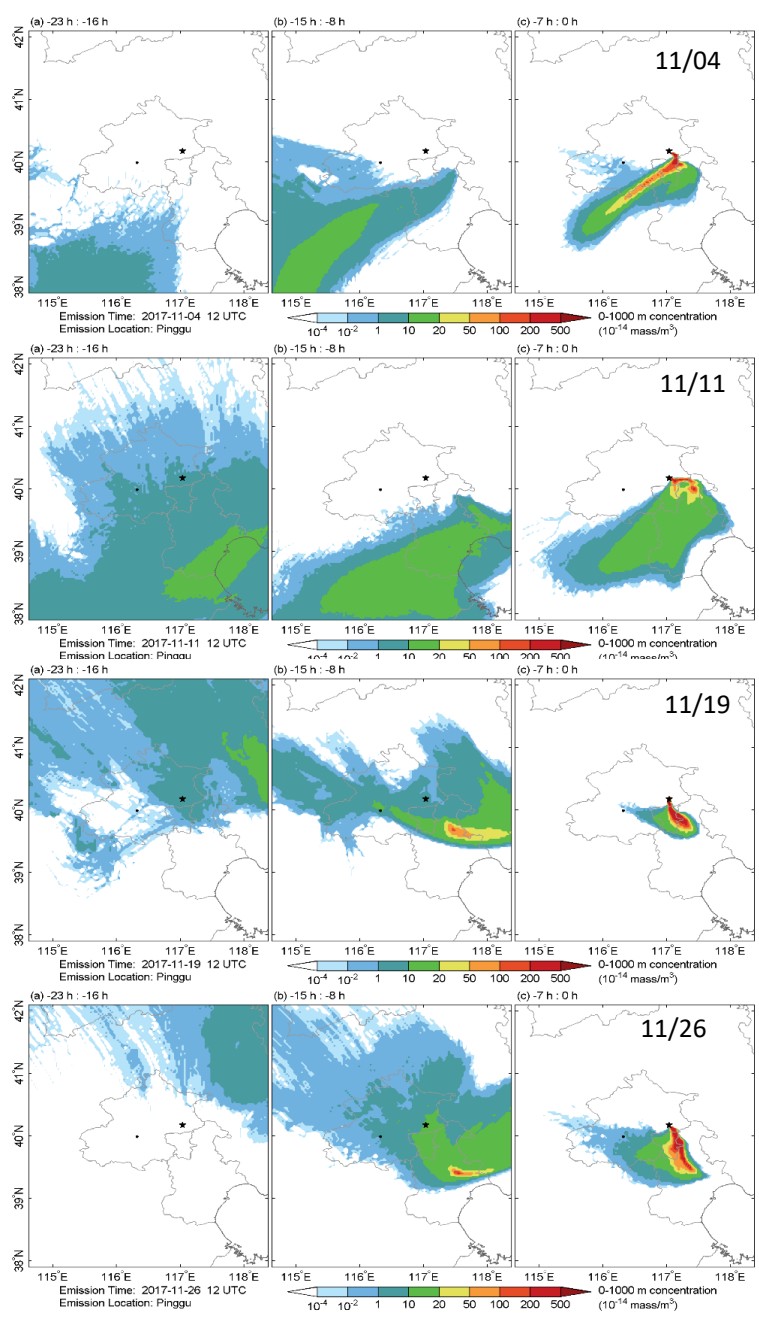


Figure 9. Typical dispersion of air mass from PG (star, on the right) to PKU (dot, on the

left) during E1 (11/04), E2 (11/11), E3 (11/19) and E4 (11/26).



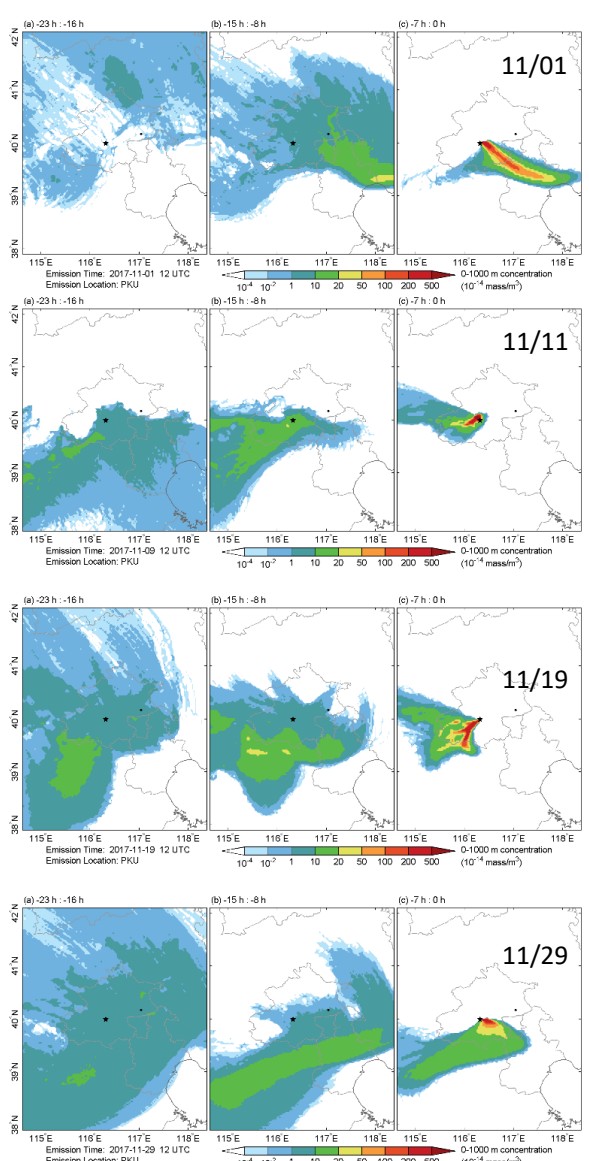

Figure 10. Typical dispersion of air mass from PKU (star, on the left) to PG (dot, on the right) in E1 (11/01), E2 (11/11), E3 (11/19) and E4 (11/29).





**3.7 Implications**
This study provides the polar plots that are used to explain the interaction of pollutants and
wind. Such regional pollution sources as BB and the coal and steel industries have a strong
impact on the particulate chemical composition of the air in urban Beijing. Besides,
according to model studies, air pollutants in such provinces as Hebei, Henan, and Shandong
are transported to Beijing (Shi et al., 2019; Du et al., 2019). In these provinces, efforts have
been made to abate emissions from the steel industry, power plants, and traffic. However,
BB accounted for 10–20% of the $PM_{2.5}$ in the study period (Liu et al., 2019). In particular,
household biofuel combustion is a primary BB source during winter, impacting both
outdoor and indoor air quality (Zhang and Cao, 2015). Therefore, more attention should be
paid to tackling BB emissions.
This study improves our general understanding of the sources of sulfates in Beijing.
Particles that only increased with the uptake of sulfate, such as OC-Sul_PKU, K-Sul_PKU,
and NaK-Sul_PKU, were transported regionally and arrived at the sampling site during
high wind speeds (> 4 m s$^{-1}$). The results are consistent with the findings of Duan et al.
(2019) that sulfates in Beijing during winter are formed regionally. Nitrate-containing
particles could be found after processing in the $NO_x$-rich urban and rural plumes of Beijing.
Since SPAMS is limited in tracking such partial organics as hydrocarbons and PAHs, the
evolution of secondary organics is unavailable in this study.
Just as Zhong et al. (2017) reported, this study found that there was a process of particle
transport before severe haze events began in Beijing. However, there are still unresolved
issues regarding the causal relationship between particle transport and haze events. There
are two possibilities. The first is that transported $PM_{2.5}$ can trigger an anomalous inversion





before a pollution event, resulting in unfavorable meteorological conditions. The second is
that transportation is a consequence of weakening atmospheric circulation causing air
stagnation. In a most recent study of aerosol–radiation feedback deterioration in Beijing
during wintertime, Wu et al. (2019) have proposed that the increase of near-surface $PM_{2.5}$
from 10 to 200 µg m$^{-3}$ can result in a decreasing of planetary boundary layer (PBL) from
1500 m to 400 m, consequently contributing the $PM_{2.5}$ concentration by 20%. They also
proposed that the wind speed decreased by 0.2 m s$^{-1}$ when the $PM_{2.5}$ loading increase from
10 to 200 m s$^{-1}$. Therefore, the southerly transported particles were impossibly to trigger
severe haze pollution due to air stagnation; the particles from both southerly transport and
accumulation were due to the attenuated near-surface atmospheric circulation.
**4. Summary**
The wintertime haze events that occurred in Beijing from 11/01/2016 to 11/29/2016 have
been investigated. The heating period, including central and residential heating in both
urban and rural areas, severely impacted the particulate chemical composition in the region.
In Beijing, a pattern of the transport and accumulation of particles was found in both the
urban and rural areas. The input of regional particles was a consequence of weakening
atmospheric circulations, resulting in the stagnation of the air which provided favorable
conditions for the accumulation of pollutants, ultimately leading to severe haze events. In
the rural area, the heavy haze was mainly controlled by air stagnation and local emissions,
but regional transport was also observed before the event. We also discussed the influence
of regional transport using the dispersion model. The air masses between PKU and PG
interacted with each other whenever heavy haze occurred. Parts I and II of this study are





useful for understanding the formation mechanism of winter haze in both the urban and
rural areas of Beijing. This study also implies that the mitigation of PM relies on both urban
and rural areas.
*Data availability.* All the data described in this study is available upon request from the
corresponding authors.
*Author contributions.* FY, MZ, TZ, and KH designed the experiments; YC, JC, ZW, MT,
CP, and HY carried them out; XY, GS, and SZ analyzed the experimental data; YC
prepared the manuscript with contributions from all coauthors.
*Competing interests.* The authors declare that they have no conflicts of interest.
*Acknowledgments.* We are grateful for financial support from the National Natural Science
Foundation of China (Grant No. 41703136 and 81571130100).



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
