# Peer review of "Simultaneous Measurement of Urban and Rural Particles in Beijing, Part II: Case Studies"

_Atmospheric Chemistry and Physics, 2019_

## Referee Comment (RC1) · Anonymous Referee #1 · 11 Mar 2020

Online single-particle chemical composition analysis was used as a tracer system to investigate the impact of heating activities and the formation of haze events in two parallel field studies at both urban and rural sites in Beijing. This manuscript focuses on case studies. One of the key points of this manuscript is that there is a pattern of transportation and accumulation of particles in both the urban and rural areas. The input of regional particles was a consequence of weakening atmospheric circulations, resulting in the stagnation of the air which provided favorable conditions for the accumulation of pollutants, ultimately leading to severe haze events. In the rural area, the heavy haze was mainly controlled by air stagnation and local emissions, but regional transport was also observed before the event.

[Figure]

This work represents a potentially substantial contribution to understanding the heavy haze formation in Beijing. However, I do have several concerns mostly related to this point. I will support the publication of this manuscript if the authors can properly address my following comments.

The hypothesis of regional transport can trigger a high pollution event is interesting. The evidence provided in this manuscript, however, can only suggest that there is some possibility of this happening at best. The evolution of particle compositions and fractions were not consistent or repeatable in the high PM events. The transition of particle fraction was not there for most of the cases. At least they are not obvious from the time series. The higher wind speed was used as another evidence for regional transport, which is far from robust. The authors did not discuss other possible causes, such as boundary layer height. The argument of why higher PM causes stagnation of atmosphere is also lacking. I wonder if the authors can provide some reasoning from the meteorological perspective. In order to support the current conclusion in the manuscript, the authors need to better illustrate those points with more than time series and address the reproducibility in all the high PM events. Otherwise, the authors can tighten their language and only provide this as one of the probable theories.

Minor Comments:

1. Line 110: Adding the two sites on this map would be helpful. 2. Line 138: Another Table 1? And it is unclear to me what these correlations are. 3. Line 273-274: " These results are consistent with the analysis of particle categories." Please expand and support this argument.

Editorial Comments: It seems that the manuscript has many typoes and I am only listing the ones I caught. Please proofread intensively before considering resubmission.

1. Line 22-33: I strongly recommend NOT to use abbreviations for particles types like EC-Nit, EC-Nit-Sul, ECOC-Nit-Sul, Nak-Nit, and OC-SulECOC-Nit in the abstract. PKU and PG have not been introduced either. Please fix. 2. Line 56-57: "Sun et al.

(2014); Sun et al.(2013a)" Format of citation needs correction. 3. Line 113: " Particle types, their ratios at both sites," Do you mean fractions instead of ratios? 4. Line 175: " control emissions from household emissions" fix typo. 5. Line 292: "such provinces as Hebei, Henan," Check grammar.

―――――――――――――――――――――

---

## Referee Comment (RC2) · Anonymous Referee #2 · 1 Apr 2020

The authors conducted two parallel studies at both urban and rural sites of Beijing, using the single-particle chemical composition as a tracing system to investigate the impact of heating activities and formation of haze events in the region. The authors argued that different types of particles were emitted between the urban central heating supply and the residential heating in the countryside. Interestingly, the authors proposed a hypothesis that the regional transport of particles could trigger heavy pollution. The study pictured the interactions of pollutants between and urban and rural sites. The reviewer recommends publications when the following concerns are addressed. Major Issues: 1. Introduction. Can the authors describe the difference between the bulk and single-particle analyses? Are there any advantages of this work compared to those

in the literature? The authors need to provide detailed information on the aim of the study. 2. The reviewer suggests analyzing the pollution events 2,3 and 4 because the detailed temporal trends are available; the missing datain E1 make them not soild. 3. Does any evidence suggest the OC-Nit-Sul formed locally? 4. The statement that the regional transport can trigger the pollution event should be clarified. The connection between wind speed with the events is not conclusive enough. Is there any additional evidence that could be provided to make a solid decision? If not, the statement could be strained base on the current data. 5. The Implication part is more like a Discussion; please consider changing that. Minor issues 1. Typos need to be checked carefully, a proofread is need for revision. 2. Through the manuscript, terms like "fractions" and "ratios" are both used. Are they the same meaning? Please clarify and be consistent if possible. 3. Line 116, please tidy Table 1, which is messy at the end. 4. Line 138, why is there two Table 1? Please check. 5. Line 175: " control emissions from household emissions" fix the typo, please reword. 6. Figures 9 and 10, please label the Events on each panel. 7. Careful proofreading is needed.

---

## Author Comment (AC1) · 31 May 2020

Dear reviewer,

We are grateful for your valuable comments that have helped us to improve our manuscript. We went through your comments and prepared a point-to-point response below. Correspondingly, the changes are also highlighted in the manuscript. All our responses are marked in blue or with "Ans". Again, we appreciate your time and comments.

Online single-particle chemical composition analysis was used as a tracer system to

investigate the impact of heating activities and the formation of haze events in two parallel field studies at both urban and rural sites in Beijing. This manuscript focuses on case studies. One of the key points of this manuscript is that there is a pattern of transportation and accumulation of particles in both the urban and rural areas. The input of regional particles was a consequence of weakening atmospheric circulations, resulting in the stagnation of the air which provided favorable conditions for the accumulation of pollutants, ultimately leading to severe haze events. In the rural area, the heavy haze was mainly controlled by air stagnation and local emissions, but regional transport was also observed before the event.

This work represents a potentially substantial contribution to understanding the heavy haze formation in Beijing. However, I do have several concerns mostly related to this point. I will support the publication of this manuscript if the authors can properly address my following comments. The hypothesis of regional transport can trigger a high pollution event is interesting. The evidence provided in this manuscript, however, can only suggest that there is some possibility of this happening at best.

The evolution of particle compositions and fractions were not consistent or repeatable in the high PM events. The transition of particle fraction was not there for most of the cases. At least they are not obvious from the time series. The higher wind speed was used as another evidence for regional transport, which is far from robust. The authors did not discuss other possible causes, such as boundary layer height. The argument of why higher PM causes stagnation of atmosphere is also lacking. I wonder if the authors can provide some reasoning from the meteorological perspective. In order to support the current conclusion in the manuscript, the authors need to better illustrate those points with more than time series and address the reproducibility in all the high PM events. Otherwise, the authors can tighten their language and only provide this as one of the probable theories.

Ans:

We are very grateful for your detailed comments. This study is based on two field measurements, and we separated our presentation into two parts. Part I illustrates the particle types, single-particle chemical composition, mixing state, and sources (Chen et al., 2020); and Part II discusses pollution events and interactions between PG and PKU.

In Part I, we investigate mass spectra of major particle types. Particle types were more aged with higher relative abundances of secondary species. Hence, we determined the aged particles from the two datasets from both PKU and PG. Considering the particles can undergo aging locally or regionally, we also studied the evolution of these particles under variable meteorological conditions. The locally aged particles arrived at sampling sites with no unique wind directions, at low wind speed (commonly < 2 m s‒1); on the other hand, regionally aged particles responded to unique wind directions. Hence, at each site, particle types can be categorized as "local" or "regional" depending on their chemical composition, hourly number counts, and wind-direction responses. The meteorological perspective on pollution events has been fully described in Part I.

In Part II, the time series of aged particles along with high wind speed has been used as indicators of regional transport. It is regrettable that we do not emphasize the time series were from the aged particles. A particle transportation event is defined if the concentrations of aged particles rose at high wind speed. For the better presentation of this part, we have added the necessary descriptions of particle evolution, transport, and meteorology-dependent properties (lines 125-144).

"We observed five particle categories at both sites: elemental carbon (EC), organic carbon (OC), internal-mixed EC and OC (ECOC), potassium-rich (K-rich), and metals. According to their different stages of atmospheric processing, the five categories can be divided into up to 20 particle types, as shown in Table 1. Particles with relative peak areas of sulfate and nitrate greater than 0.1 were marked with nitrate (-Nit) or sulfate (-Sul), respectively, or both (-Nit-Sul). Particle types were more aged with higher relative abundances of secondary species. Besides, the suffixes "_PKU" and "_PG" are used

when the same particles appear. The typical single-particle mass spectra of all particle types are available in Supportive Information and (Chen et al., 2020).

After resolving the sources, the origins of particle types were ascertained at both sites. At PKU, the following particle types were local: EC-Nit, EC-Nit-Sul, ECOC-Nit-Sul, Ca-rich, and ECOC-Nit. These particles arrived at PKU with no unique wind directions, at low wind speed (commonly < 2 m s‒1) and with clear diurnal patterns. On the contrary, OC-Nit, OC-Sul, NaK-Nit, and NaK-Nit-Sul responded to unique wind directions, implying that these particle types were regionally transported. At PG, all particle types showed patterns that were both local and regional. For example, OC, ECOC, OC-Nit-Sul, and ECOC-Nit-Sul came from the local area, northeast, and southwest. Universal patterns can be used to determine the mechanisms of pollution event formation when combined with unique cases."

We also discussed the shift of PBL and provided other evidence to support our hypothesis (lines 252-263). "In the most recent study of aerosol-radiation feedback deterioration in Beijing during wintertime, Wu et al. (2019) proposed that the increase of near-surface PM2.5 from 10 to 200 $\mu$g m‒3 can result in a decrease of the planetary boundary layer (PBL) from 1,500 m to 400 m, consequently contributing to PM2.5 concentration by 20%. However, a 20% difference cannot explain that PM2.5 concentration increased from 100 $\mu$g m‒3 to 300 $\mu$g m‒3. Moreover, when PM2.5 exceeded 200 $\mu$g m‒3, the height of the PBL remained at 400‒500 m, and air stagnation occurred with weak horizontal wind and inactive advection. Zhong et al. (2017) observed that weak temperature inversion occurred almost at the same period, and near-surface RH increased after southerly transport, along with decreased vertical wind speed and increased RH during winter. Air stagnation was also observed in this study (Figure 2). Therefore, based on the evidence of chemical evolution, the southerly transport of PM strongly connected to pollution events. "

Transported aerosols weaken near-surface radiation, causing the shifting of the PBL with temperature inversion and increasing RH. The consequences are favorable for accumulation and secondary formation, which were both observed in this study. There-
fore, the claim that the regional transport of pollutants is highly linked to pollution events
is feasible.

Minor Comments:

1. Line 110: Adding the two sites on this map would be helpful.

Ans: A map of the sampling site has been added in line 101.

2. Line 138: Another Table 1? And it is unclear to me what these correlations are.

Ans: We are sorry, it was a typo, and we have fixed it (line 167).

3. Line 273-274: "These results are consistent with the analysis of particle categories."
Please expand and support this argument.

Ans: Yes, the argument is incomplete. We have added the following evidence to sup-
port our statement (lines 324‒327): "As shown in Figure 3, when transport occurred
on November 4th, 19th, and 26th, regional particle types such as K-Nit-Sul, Nak-Nit-
Sul, ECOC-Nit-Sul, and OC-Nit-Sul increased due to transport from the east (Part I).
"

Editorial Comments: It seems that the manuscript has many typos and I am only listing
the ones I caught. Please proofread intensively before considering resubmission.

Ans:

Thank you for the reminder. A full proofread has been conducted by a native English-
speaking scientist.

1. Line 22-33: I strongly recommend NOT to use abbreviations for particles types
like EC-Nit, EC-Nit-Sul, ECOC-Nit-Sul, Nak-Nit, and OC-SulECOC-Nit in the abstract.
PKU and PG have not been introduced either. Please fix. Ans: We have added the
necessary information on those terms (Lines 96–99).

2. Line 56-57: "Sun et al. (2014); Sun et al.(2013a)" Format of citation needs correction. Ans: The citation format has been updated to "Sun et al. (2013) and (2014)".

3. Line 113: "Particle types, their ratios at both sites," Do you mean fractions instead of ratios? Ans: Yes, we mean their number fractions in the dataset. We have changed all terms to "fractions."

4. Line 175: "control emissions from household emissions" fix typo. Ans: This has been changed to (lines 205-206): "Conclusively, the control of emissions from household heating is also key to improving the air quality in Beijing."

5. Line 292: "such provinces as Hebei, Henan," Check grammar Ans:

This has been changed to (lines 344): "...air pollutants in Hebei, Henan, and Shandong provinces are transported to Beijing (Shi et al., 2019; Du et al., 2019)."

References

Chen, Y., Cai, J., Wang, Z., Peng, C., Yao, X., Tian, M., Han, Y., Shi, G., Shi, Z., Liu, Y., Yang, X., Zheng, M.,

Zhu, T., He, K., Zhang, Q., and Yang, F.: Simultaneous Measurement of Urban and Rural Single Particles in Beijing, Part I: Chemical Composition and Mixing State, Atmos. Chem. Phys. Discuss., 2020, 1-40, 10.5194/acp-2019-933, 2020. Du, H., Li, J., Chen, X., Wang, Z., Sun, Y., Fu, P., Li, J., Gao, J., and Wei, Y.: Modeling of aerosol property evolution during winter haze episodes over a megacity cluster in northern China: roles of regional transport and heterogeneous reactions of SO2, Atmos. Chem. Phys., 19, 9351-9370, 10.5194/acp-19-9351-2019, 2019.

Shi, Z., Vu, T., Kotthaus, S., Harrison, R. M., Grimmond, S., Yue, S., Zhu, T., Lee, J., Han, Y., Demuzere, M., Dunmore, R. E., Ren, L., Liu, D., Wang, Y., Wild, O., Allan, J., Acton, W. J., Barlow, J., Barratt, B., Beddows, D., Bloss, W. J., Calzolai, G., Carruthers, D., Carslaw, D. C., Chan, Q., Chatzidiakou, L., Chen, Y., Crilley, L., Coe, H., Dai, T., Doherty, R., Duan, F., Fu, P., Ge, B., Ge, M., Guan, D., Hamilton, J. F., He, K., Heal,

M., Heard, D., Hewitt, C. N., Hollaway, M., Hu, M., Ji, D., Jiang, X., Jones, R., Kalberer, M., Kelly, F. J., Kramer, L., Langford, B., Lin, C., Lewis, A. C., Li, J., Li, W., Liu, H., Liu, J., Loh, M., Lu, K., Lucarelli, F., Mann, G., McFiggans, G., Miller, M. R., Mills, G., Monk, P., Nemitz, E., O'Connor, F., Ouyang, B., Palmer, P. I., Percival, C., Popoola, O., Reeves, C., Rickard, A. R., Shao, L., Shi, G., Spracklen, D., Stevenson, D., Sun, Y., Sun, Z., Tao, S., Tong, S., Wang, Q., Wang, W., Wang, X., Wang, X., Wang, Z., Wei, L., Whalley, L., Wu, X., Wu, Z., Xie, P., Yang, F., Zhang, Q., Zhang, Y., Zhang, Y., and Zheng, M.: Introduction to the special issue "In-depth study of air pollution sources and processes within Beijing and its surrounding region (APHH-Beijing)", Atmos. Chem. Phys., 19, 7519-7546, 10.5194/acp-19-7519-2019, 2019.

Wu, J., Bei, N., Hu, B., Liu, S., Zhou, M., Wang, Q., Li, X., Liu, L., Feng, T., Liu, Z., Wang, Y., Cao, J., Tie, X.,

Wang, J., Molina, L. T., and Li, G.: Aerosol–radiation feedback deteriorates the wintertime haze in the North China Plain, Atmos. Chem. Phys., 19, 8703-8719, 10.5194/acp-19-8703-2019, 2019.

Zhong, J., Zhang, X., Wang, Y., Sun, J., Zhang, Y., Wang, J., Tan, K., Shen, X., Che, H., Zhang, L., Zhang, Z., Qi, X., Zhao, H., Ren, S., and Li, Y.: Relative contributions of boundary-layer meteorological factors to the explosive growth of PM2.5 during the red-alert heavy pollution episodes in Beijing in December 2016, Journal of Meteorological Research, 31, 809-819, 10.1007/s13351-017-7088-0, 2017.

Please also note the supplement to this comment:
https://www.atmos-chem-phys-discuss.net/acp-2019-1118/acp-2019-1118-AC1-supplement.pdf

**Supplement:**

*Supporting information of*

Simultaneous Measurement of Urban and Rural Particles in Beijing, Part II: Case Studies of Haze Events and Regional Transport

Yang Chen,[1] Guangming Shi,[1, 3] Jing Cai,[2] Zongbo Shi,[4, 5] Zhichao Wang,[1] Xiaojiang Yao,[1] Mi Tian,[1] Chao Peng,[1] Yiqun Han,[2] Tong Zhu,[2] Yue Liu,[2] Xi Yang,[2] Mei Zheng,[2*] Fumo Yang,[1, 3*] and Kebin He[6]

[1] Chongqing Institute of Green and Intelligent Technology, Chinese Academy of Sciences, Chongqing 400714, China

[2] SKL-ESPC and BIC-ESAT, College of Environmental Sciences and Engineering, Peking University, Beijing 100871, China

[3] Department of Environmental Science and Engineering, College of Architecture and Environment, Sichuan University, Chengdu 610065, China

[4] School of Geography, Earth and Environmental Sciences, the University of Birmingham, Birmingham B15 2TT, UK

[5] Institute of Surface-Earth System Science, Tianjin University, Tianjin 300072, China

[6] School of Environment, Tsinghua University, Beijing 100084, China

**Methodology**

**Sampling sites**

The campaigns were performed simultaneously at PKU (116.32ºE, 39.99ºN) and PG (117.05ºE, 40.17ºN) from 11/01/2016 to 11/29/2016. A Description of the PKU site is available in the literature (Huang et al., 2006). Briefly, the site is located on the rooftop (15 m above the ground) on the PKU campus which is surrounded by residential and commercial blocks. Trace gases (Thermo Inc. series), meteorological parameters (Vaisala Inc.), and $PM_{2.5}$ (TEOM 1430) were recorded during the observation.

The PG site (117.053ºE, 40.173ºN) is 3 km from the PG center. The site is located in the northeast of the PKU site with a distance of 70 km. The PG site also acts as a host of the AIRLESS (Effects of AIR pollution on the cardiopulmonary disease in urban and peri-urban residents in Beijing) Project. The meteorological data is acquired from the local meteorological office. The PG village is surrounded by orchards and farmland with no main road nearby on a scale of 3 km. Coal and biomass are used for domestic heating and cooking in the nearby villages.

**Instrumentation and data analysis**

Two SPAMSs (Model 0515, Hexin Inc., Guangzhou, China) were deployed at both PKU and PG. A technical description of SPAMS is available in (Li et al., 2011). Briefly, a SPAMS has three functional parts: sampling, sizing, and mass spectrometry. In the sampling part, particles within a 0.1–2.0 µm size range pass efficiently through an aerodynamic lens. In the sizing unit, the aerodynamic diameter ($D_{va}$) is calculated using

the time-of-flight of particles. The particles are then decomposed and ionized into ions one-by-one using a 266 nm laser. A bipolar time-of-flight mass spectrometer measures the ions and generates the positive and negative mass spectra of each particle. The two instruments were maintained and calibrated following the standard procedures before sampling (Chen et al., 2017).

A neural network algorithm based on adaptive resonance theory (ART-2a) was used to resolve particle types from both datasets (Song et al., 1999). The parameters used were: a vigilance factor of 0.70, a learning rate of 0.05, and 20 iterations. This procedure generated 771 and 792 particle groups. Then, the groups were combined into particle types based on similar mass spectra, temporal trends, and size distributions (Dallosto and Harrison, 2006). During combining, relative areas of nitrate and sulfate were used to distinguish the stages of processing, assuming that more sulfate and nitrate can be measured if a particle is more processed during its lifetime. Thus, particles with relative peak areas of sulfate and nitrate larger than 0.1 were marked with nitrate (-Nit), sulfate (-Sul), respectively, or both. Finally, the strategy resulted in 20 and 19 particle types at PKU and PG respectively. Among them, 17 types appeared at both sites, and each type has identical mass spectra ($R^2 > 0.80$) between each other.

**Paritcle types overview**

Table S1. Relative abundance of particle types in polluted and clear days at the PKU site

|  | E1 | Clear1 | E2 | Clear2 | E3 | Clear2 |
|---|---|---|---|---|---|---|
| BB_PKU | 0.05 | 0.08 | 0.05 | 0.17 | 0.07 | 0.17 |
| CA _PKU | 0.00 | 0.00 | 0.01 | 0.00 | 0.00 | 0.00 |
| EC-Nit_PKU | 0.08 | 0.08 | 0.10 | 0.01 | 0.02 | 0.01 |
| EC-Nit-Sul_PKU | 0.12 | 0.09 | 0.13 | 0.05 | 0.08 | 0.05 |
| EC-Sul_PKU | 0.00 | 0.01 | 0.01 | 0.04 | 0.01 | 0.04 |
| ECOC-Nit_PKU | 0.03 | 0.04 | 0.03 | 0.01 | 0.02 | 0.01 |
| ECOC-Nit-Sul_PKU | 0.10 | 0.11 | 0.13 | 0.08 | 0.18 | 0.08 |
| ECOC-Sul_PKU | 0.06 | 0.14 | 0.13 | 0.28 | 0.18 | 0.28 |
| Fe-rich_PKU | 0.03 | 0.02 | 0.03 | 0.01 | 0.02 | 0.01 |
| K-Amin-Nit_PKU | 0.00 | 0.00 | 0.00 | 0.01 | 0.00 | 0.01 |
| KAS_PKU | 0.11 | 0.07 | 0.08 | 0.02 | 0.04 | 0.02 |
| Ksec-Nit-Sul_PKU | 0.22 | 0.20 | 0.12 | 0.12 | 0.12 | 0.12 |
| Ksec-Sul_PKU | 0.01 | 0.01 | 0.00 | 0.02 | 0.01 | 0.02 |
| NaK_PKU | 0.00 | 0.00 | 0.00 | 0.01 | 0.00 | 0.01 |
| NaK-Nit_PKU | 0.07 | 0.03 | 0.06 | 0.04 | 0.08 | 0.04 |
| NaK-Nit-Sul_PKU | 0.02 | 0.01 | 0.02 | 0.02 | 0.06 | 0.02 |
| NaK-Sul_PKU | 0.00 | 0.00 | 0.00 | 0.01 | 0.00 | 0.01 |
| OC-Nit_PKU | 0.01 | 0.01 | 0.01 | 0.01 | 0.01 | 0.01 |
| OC-Nit-Sul_PKU | 0.06 | 0.08 | 0.08 | 0.07 | 0.09 | 0.07 |
| OC-Sul_PKU | 0.01 | 0.01 | 0.01 | 0.02 | 0.01 | 0.02 |

Table S2. Relative abundance of particle types in polluted and clear days at the PG site

| | E1 | CLEAR1 | E2 | CLEAR2 | E3 | E4 | CLEAR4 |
|---|---|---|---|---|---|---|---|
| BB_PG | 0.07 | 0.10 | 0.06 | 0.10 | 0.04 | 0.07 | 0.11 |
| Ca-rich_PG | 0.00 | 0.00 | 0.00 | 0.00 | 0.00 | 0.00 | 0.00 |
| EC-Nit_PG | 0.02 | 0.01 | 0.04 | 0.00 | 0.02 | 0.00 | 0.00 |
| EC-Nit-Sul_PG | 0.05 | 0.02 | 0.05 | 0.01 | 0.03 | 0.02 | 0.01 |
| ECOC-Nit-Sul_PG | 0.04 | 0.09 | 0.06 | 0.09 | 0.05 | 0.06 | 0.11 |
| EC-Sul_PG | 0.00 | 0.00 | 0.00 | 0.00 | 0.00 | 0.00 | 0.00 |
| ECOC-Nit-Sul_PG | 0.15 | 0.12 | 0.19 | 0.10 | 0.21 | 0.22 | 0.12 |
| ECOC-Sul_PG | 0.06 | 0.15 | 0.10 | 0.30 | 0.11 | 0.07 | 0.14 |
| Fe-rich_PG | 0.03 | 0.00 | 0.02 | 0.00 | 0.03 | 0.01 | 0.00 |
| Ksec-Nit_PG | 0.06 | 0.04 | 0.04 | 0.02 | 0.04 | 0.03 | 0.03 |
| Ksec_Nit-Sul_PG | 0.03 | 0.02 | 0.02 | 0.01 | 0.02 | 0.02 | 0.01 |
| Ksec_PG | 0.05 | 0.05 | 0.04 | 0.05 | 0.04 | 0.03 | 0.05 |
| Ksec-Sul_PG | 0.07 | 0.08 | 0.03 | 0.04 | 0.04 | 0.04 | 0.03 |
| Nak-Nit_PG | 0.02 | 0.01 | 0.01 | 0.00 | 0.01 | 0.03 | 0.01 |
| Nak-Nit-Sul_PG | 0.02 | 0.01 | 0.01 | 0.00 | 0.01 | 0.05 | 0.01 |
| NaK_PG | 0.02 | 0.03 | 0.02 | 0.02 | 0.01 | 0.02 | 0.03 |
| NaK-Sul_PG | 0.00 | 0.01 | 0.01 | 0.00 | 0.00 | 0.00 | 0.00 |
| OC-Nit-Sul_PG | 0.22 | 0.17 | 0.22 | 0.13 | 0.24 | 0.23 | 0.18 |
| OC_PG | 0.03 | 0.03 | 0.04 | 0.03 | 0.04 | 0.03 | 0.04 |
| OC-Sul_PG | 0.05 | 0.07 | 0.05 | 0.09 | 0.07 | 0.08 | 0.12 |

Table S3. Correlations of number fractions of particle types at PG.

|        | E1   | CLEAR1 | E2   | CLEAR2 | E3   | E4   | CLEAR4 |
|--------|------|--------|------|--------|------|------|--------|
| E1     | 1.00 |        |      |        |      |      |        |
| CLEAR1 | 0.82 | 1.00   |      |        |      |      |        |
| E2     | 0.95 | 0.85   | 1.00 |        |      |      |        |
| CLEAR2 | 0.50 | 0.86   | 0.62 | 1.00   |      |      |        |
| E3     | 0.95 | 0.85   | 0.99 | 0.62   | 1.00 |      |        |
| E4     | 0.93 | 0.82   | 0.95 | 0.54   | 0.96 | 1.00 |        |
| CLEAR4 | 0.77 | 0.95   | 0.83 | 0.84   | 0.83 | 0.82 | 1.00   |

**Mass Spectra**

[Figure]

Figure S1. Unique Single particle mass spectra of particle types at PKU.

[Figure]

Figure S2. Single particle mass spectra of OC-related particle types at both PKU and PG.

[Figure]

Figure S3. Single particle mass spectra of NaK-related particle types at both PKU and PG.

[Figure]

Figure S4. Single particle mass spectra of KSecondary-related particle types at both PKU and PG.

[Figure]

Figure S5. Single particle mass spectra of ECOC-related particle types at both PKU and PG.

[Figure]

Figure S6. Single particle mass spectra of EC-related particle types at both PKU and PG.

[Figure]

Figure S7. Single particle mass spectra of metal-related particle types at both PKU and PG.

[Figure]

Figure S8. Single particle mass spectra of Nitrate-rich particle types at PKU.

[Figure]

Figure S9. Normalized ratios of particle types at PKU during three pollution events.

[Figure]

Figure S10. Normalized ratios of particle types at PG during four pollution events.

[Figure]

Figure S11. The relative abundance of normalized number ration of particle types in the pollution events and adjacent clear days.

[Figure]

Figure S11. Air mass dispersed from PG (star, on the right) to PKU (dot, on the left) during the observation period.

[Figure]

Figure S12. Air mass dispersed from PKU (star, on the left) to PG (dot, on the right) during the observation period.

---

## Author Comment (AC2) · 31 May 2020

Dear reviewer,

We are very grateful for your review of our manuscript. We appreciate your comments and suggestions, and they are precious for us in improving this study. We have prepared a detailed point-by-point response highlighted in blue or "Ans."

Chen et al., The authors conducted two parallel studies at both urban and rural sites of Beijing, using the single-particle chemical composition as a tracing system to investigate the impact of heating activities and formation of haze events in the region.

[Figure]

The authors argued that different types of particles were emitted between the urban central heating supply and the residential heating in the countryside. Interestingly, the authors proposed a hypothesis that the regional transport of particles could trigger heavy pollution. The study pictured the interactions of pollutants between and urban and rural sites. The reviewer recommends publications when the following concerns are addressed. We appreciate your positive comments.

Major Issues:

1. Introduction. Can the authors describe the difference between the bulk and single-particle analyses? Are there any advantages of this work compared to those in the literature? The authors need to provide detailed information on the aim of the study.

Ans: We have enhanced the related literature review (lines 71-76):

"SPAMS has proven a useful tool for characterizing the single-particle chemical composition, mixing state, and processing of atmospheric particles (Chen et al., 2019a). Single-particle chemical composition and mixing state can be used as a tracing system to explore the sources and origins of unique particle types (Chen et al., 2019b; Li et al., 2016). For example, by combining meteorological parameters, we can determine the sources and transport conditions of specific particle types (Chen et al., 2018; Chen et al., 2020). "

The aims of the study have been expanded (lines 77-83):

"As mentioned in Part I (Chen et al., 2020), two SPAMSs were deployed simultaneously at Peking University (PKU) and Pinggu (PG) to monitor urban and rural particles in the Beijing region. In Part II, the resolved particle types are used to trace the evolution, transport, and formation of pollution events. The detailed analysis of haze events and effects of heating activities are addressed. Combining field measurements and model studies, the interactions between the two sampling sites, representing urban and rural eastern areas, are systematically analyzed."
2. The reviewer suggests analyzing the pollution events 2,3 and 4 because the detailed temporal trends are available; the missing information in E1 could misleading.

Ans: Do you mean the pollution event at PG? Yes, there is a 10 h gap in the temporal trends in E1_PG, but this gap did not influence our analysis of particulate pollution. As shown in Figure 8, the temporal trends of major particle types were clear, and patterns can be found.

3. Sections 3.4 and 3.5. There is too much description in a single site, which causes difficulties in comparing the patterns of the pollution events. The reviewer strongly suggests illustrating E1 at both PKU and PG together, making the comparison much easier than the current form.

Ans: We appreciate the comments and have added a supplementary statement to the text (lines 284-292): "Both E1_PG and E1_PKU had patterns of transport and accumulation, but the transported particles were different; for example, at the PG site, the appearance of EC-Nit and EC-Nit-Sul, which came from the west, i.e., urban Beijing, was pronounced, while at PKU, particle types such as OC-Nit-Sul, K-Nit-Sul, K-Nit, NaK-Nit, and K-Nit-Sul increased dramatically due to transport. These particle types were emitted from residential heating in rural areas. In the accumulation stages at both sites, the concentrations of local particles rose, such as EC-Nit-Sul at PKU and NaK-Nit-Sul at PG. In short, the evolution of particles, including both transport and accumulation at both PKU and PG, were affected by the movement of air mass and local emissions."

4. Does any evidence suggest the OC-Nit-Sul formed locally? Ans: As we proposed in Part I (Chen et al., 2020), OC-Nit-Sul_PKU had two origins. One was local formation during air stagnation along with no unique wind direction; the other was from the southwest.

5. The statement that the regional transport can trigger the pollution event should be clarified. The connection between wind speed with the events is not conclusive enough.

Is there any additional evidence that could be provided to make a solid decision? If not, the statement could be strained base on the current data.

Ans: Please refer to Sections 3.1 and 3.4. We have discussed the influence of meteorological parameters and aerosol-radiation feedback. Combining the results from multiple studies, we conclude that. . .

6. The Implication part is more like a Discussion; please consider changing that. Ans: We decided to leave it as it is because the two paragraphs are a summary of what we need to mention after long Results and Discussion sections. But we did move the discussion of aerosol-radiation feedback and the evolution of PBL into Section 3.4 for greater coherence.

Minor issues

1. Typos need to be checked carefully, a proofread is need for revision.

Ans: Fixed.

2. Through the manuscript, terms like "fractions" and "ratios" are both used. Are they the same meaning? Please clarify and be consistent if possible.

Ans: We have uniformly used "fractions" to illustrate the relative changes of particle types. Thank you for the reminder.

3. Line 116, please tidy Table 1, which is messy at the end.

Ans: Fixed.

4. Line 138, why is there two Table 1? Please check. Ans: It was a typo, and we have fixed it.

5. Line 175: " control emissions from household emissions" fix the typo, please reword.

Ans: Fixed.

6. Figures 9 and 10, please label the Events on each panel.

Ans: We have added descriptions in the Figure captions.

7. Careful proofreading is needed.

Ans: Fixed.

References

Chen, Y., Liu, H., Yang, F., Zhang, S., Li, W., Shi, G., Wang, H., Tian, M., Liu, S., Huang, R., Wang, Q., Wang, P., and Cao, J.: Single particle characterization of summertime particles in Xi'an (China), Sci. Total Environ., 636, 1279-1290, 10.1016/j.scitotenv.2018.04.388, 2018.

Chen, Y., Tian, M., Huang, R.-J., Shi, G., Wang, H., Peng, C., Cao, J., Wang, Q., Zhang, S., Guo, D., Zhang, L., and Yang, F.: Characterization of urban amine-containing particles in southwestern China: seasonal variation, source, and processing, Atmos. Chem. Phys., 19, 3245-3255, 10.5194/acp-19-3245-2019, 2019b.

Chen, Y., Cai, J., Wang, Z., Peng, C., Yao, X., Tian, M., Han, Y., Shi, G., Shi, Z., Liu, Y., Yang, X., Zheng, M., Zhu, T., He, K., Zhang, Q., and Yang, F.: Simultaneous Measurement of Urban and Rural Single Particles in Beijing, Part I: Chemical Composition and Mixing State, Atmos. Chem. Phys. Discuss., 2020, 1-40, 10.5194/acp-2019-933, 2020.

Li, W., Shao, L., Zhang, D., Ro, C.-U., Hu, M., Bi, X., Geng, H., Matsuki, A., Niu, H., and Chen, J.: A review of single aerosol particle studies in the atmosphere of East Asia: morphology, mixing state, source, and heterogeneous reactions, J Clean Prod, 112, 1330-1349, 10.1016/j.jclepro.2015.04.050, 2016.